# Polymorphism of *VRTN* Gene g.20311_20312ins291 Was Associated with the Number of Ribs, Carcass Diagonal Length and Cannon Bone Circumference in Suhuai Pigs

**DOI:** 10.3390/ani10030484

**Published:** 2020-03-13

**Authors:** Nengjing Jiang, Chenxi Liu, Tingxu Lan, Qian Zhang, Yang Cao, Guang Pu, Peipei Niu, Zongping Zhang, Qiang Li, Juan Zhou, Xiaokui Li, Liming Hou, Ruihua Huang, Pinghua Li

**Affiliations:** 1Institute of Swine Science, Nanjing Agricultural University, Nanjing 210095, China; jiangnengjing@163.com (N.J.); liuchenxi0018@163.com (C.L.); ltx17681060270@163.com (T.L.); zhangqian_1992@126.com (Q.Z.); hudiebuhuifei@126.com (Y.C.); pigfarmer_pu@163.com (G.P.); mnhouliming@126.com (L.H.); rhhuang@njau.edu.cn (R.H.); 2Huaian Academy, Nanjing Agricultural University, Huaian 223005, China; niupeipei2@126.com (P.N.); zongpingzhang@126.com (Z.Z.); 3Huaiyin pig Breeding Farm of Huaian City, Huaian 223322, China; 13912078083@139.com (Q.L.); 15261762741@163.com (J.Z.); 4Guangdong Wangjiang Pig Breeding Company Limited, Yangjiang 529948, China; lixiaokui-336208@163.com

**Keywords:** *VRTN* gene g.20311_20312ins291, the number of ribs, cannon bone circumference, Suhuai pigs

## Abstract

**Simple Summary:**

An increase in the number of ribs (RIB) could improve carcass length (CL) and body size. Cannon bone circumference (CBC) is a pivotal body size trait, and a large CBC could enhance the capacity to bear excessive body weight, vigorous exercise, and resistance to injuries. Several researchers showed that the vertnin (*VRTN*) gene g.20311_20312ins291 (NC_010449.5 7: g.20311_20312ins291) is an important variant that is related to RIB and CL of Western pigs. However, it is unknown whether this variant could affect the CBC of pigs. Our study showed that this variant was significantly associated with RIB, carcass diagonal length (CDL), and CBC in Suhuai pigs; therefore, it could be used as a potential molecular marker for improving RIB, CDL, and CBC in this breed.

**Abstract:**

The vertnin (*VRTN*) gene g.20311_20312ins291 was reported as an important variant related to the number of ribs (RIB), and the ins/ins genotype was advantageous for improving RIB of Western pigs. The purpose of this study was to determine whether the *VRTN* gene g.20311_20312ins291 influences RIB, carcass traits, and body size traits, including cannon bone circumference (CBC) in Chinese Suhuai pigs. We found that the *VRTN* gene g.20311_20312ins291 was polymorphic in Suhuai fattening pigs and gilts. The polymorphism of g.20311_20312ins291 was significantly associated with RIB and CDL in Suhuai fattening pigs (*p* < 0.01), whereas this variant had no influence on carcass weight (CWT). There was a tendency of association between this variant and carcass straight length (CSL) in Suhuai fattening pigs (*p* = 0.06). The polymorphism of g.20311_20312ins291 was also significantly associated with CBC in Suhuai gilts (*p* = 0.04). Furthermore, CBC was positively genetically correlated with body length (0.22, *p* < 0.01) and body weight (0.15, *p* < 0.01). Our results indicated that the *VRTN* gene g.20311_20312ins291 could be used as a potential marker for improving RIB, CDL, and CBC in Suhuai pigs.

## 1. Introduction 

Carcass traits, such as weight and length, are of economic importance in meat production. Increasing body size traits such as body length, chest circumference, and cannon bone circumference are goals pursued in pig breeding in China. Recently, a series of studies demonstrated that the number of vertebrae is an economic trait that affects carcass length and meat production [1]. The heritability of the number of vertebrae is high (0.60–0.62), and the number of vertebrae is positively correlated with body length [2,3]. The vertebrae of pigs are classified into five parts, including cervical, thoracic, lumbar, sacral, and caudal vertebrae. The numbers of cervical vertebrae, sacral vertebrae, and caudal vertebrae of pigs are fixed at 7, 4, and 5, respectively [4]. The thoracic and lumbar vertebrae are the main components of the vertebrae, and there is a large range of variations in their number. The thoracic vertebral number (TVN) ranges from 13 to 17, and the lumbar vertebral number (LVN) ranges from 5 to 7 in Western modern breeds [2,5]. There are 19 thoracic-lumbar vertebrae in wild boar, and the total numbers of thoracic and lumbar vertebrae ranged from 19 to 20 in Chinese indigenous breeds [6]. Compared with Chinese indigenous breeds, the Western commercial breeds, including Large White, Duroc, and Landrace, have more thoracic-lumbar vertebrae (*n* = 21 to 23) owing to high-intensity selection and breeding.

From the perspective of pork consumption and the economic value of vertebrae, sparerib is one of the most valuable parts of the pork carcass. The number of ribs (RIB) is thus more important than LVN. Multiple researchers have focused on the genetic analysis of RIB. Some studies reported quantitative trait loci (QTL) mapping of vertebral numbers. The F_2_ resource population of Meishan female pigs × Göttingen miniature male pigs was used to locate the QTLs affecting the vertebrae to the position near SW705 on *Sus scrofa* chromosome 1 (SSC1) and near SW252 on *Sus scrofa* chromosome 7 (SSC7), respectively [7,8]. Subsequently, the F_2_ resource population of Large White inbred families × 11 European breeds was constructed to refine the QTL to 41kb (SJ7121-SJ7114), which only contained one gene encoding an unknown protein (C140rf195) named vertnin (*VRTN*). There were 291 bp in the first intron of the *VRTN* gene that was named PRE1-SINE element and could affect the expression of the *VRTN* gene [9]. Moreover, the *VRTN* gene was found to affect the number of vertebrae in the F_2_ resource population of White Duroc × Erhualian pigs [10]. The *VRTN* gene g.20311_20312ins291 was further confirmed to be one of the causative variants related to the TVN of Chinese Sutai pigs (Chinese Taihu × Duroc) and Western commercial hybrid (Duroc × Landrace × Large White; DLL) populations [11]. Recently, the *VRTN* gene was reported to regulate the development of thoracic vertebrae in mammals via the Notch signaling pathway. VRTN is a novel transcription factor that affects vertebral development. Half of *VRTN* heterozygous (Vrtn^+/−^) mice showed abnormal vertebral development with fewer thoracic vertebrae than wild-type littermates [12]. 

Apart from the *VRTN* gene, other candidate genes that affect vertebrae have been reported. The QTLs affecting the number of lumbar vertebrae and thoracolumbar vertebrae (TLVN) have been detected to be located in the *Hox B* gene cluster, and QTLs located in the *Hox A* gene cluster were significantly associated with TLVN [13]. A new candidate gene, *LTBP2* (latent transforming growth factor binding protein 2), was found to affect RIB in 596 Large White × Chinese Min pigs F_2_ intercrosses [14]. In addition, a missense mutation (c.4481 A > C) in the *LTBP2* gene was associated with TVN in 1105 F_2_ animals of a Landrace crossed with Korean native pigs [15]. Moreover, the 105179474 G > A loci in *TGFβ3* (transforming growth factor beta 3) gene on SSC7 was found to be associated with RIB and TLVN in the F_2_ resource population of Large White × Min pigs (*n* = 567) [16]. Cannon bone circumference (CBC) is an important body size trait, and a large CBC could enhance the capacity to bear excessive body weight, vigorous exercise, and to resist injuries. When RIB and CL of pigs increased due to selection, the body size increases, which requires a larger CBC to support the greater body weight. Therefore, it is important to select RIB and CBC of pigs, simultaneously. Although previous studies showed that the *VRTN* gene g.20311_20312ins291 is an important variant affecting RIB and CL of Western pigs, it is still unknown whether this variant is associated with CBC.

Chinese Suhuai pig is a newly cultivated breed, which contains approximately 25% Chinese Huai and 75% Large White ancestries. Suhuai pig inherits high tolerance to coarse feed from Chinese Huai pig and high lean meat yield from Large White [17,18]. At present, more than 10,000 Suhuai sows are raised in over 20 provinces in China [18]. Since Large White has more RIB and larger CBC than Chinese indigenous pigs, we speculated that there are phenotypic variations of RIB and CBC within Suhuai pig populations. Increasing RIB, CBC, and other body size traits are essential goals in the breeding program of Suhuai pigs. Therefore, this study aims to identify whether the *VRTN* gene g.20311_20312ins291 is polymorphic and has an influence on RIB, carcass traits, CBC, and other body size traits of Suhuai pigs.

## 2. Materials and Methods

### 2.1. Ethics Statement

All experiments were performed according to Guidelines for the Care and Use of Laboratory Animals prepared by the Institutional Animal Welfare and Ethics Committee of Nanjing Agricultural University, Nanjing, China. All experimental protocols were approved by the Nanjing Agricultural University Animal Care and Use Committee (Certification No.: SYXK (Su) 2017-0007).

### 2.2. Animals and Phenotype Measurement

In this study, 655 experimental animals were provided by Huaiyin breeding farm (Huaian, China), including 335 healthy Suhuai fattening pigs (SH-F) and 320 healthy Suhuai gilts (SH-G) that were raised under the same feeding and management conditions. The average weight of these Suhuai fattening pigs was 87.61 ± 0.54 kg, and these pigs were slaughtered in three batches in the Huaian Jinyuan Meat products Co., Ltd. (Huaian, China). We measured the number of ribs (RIB), carcass straight length (CSL), carcass diagonal length (CDL), and carcass weight (CWT) of SH-F. We measured the body size and body weight traits of 320 Suhuai gilts aged about 160 ± 10 days, including chest circumference (CC), abdominal circumference (AC), cannon bone circumference (CBC), rump circumference (RC), body length (BL), and body weight (BW).

### 2.3. DNA Extraction and VRTN Gene g.20311_20312ins291 Genotyping

Ear tissues from the end of the right ear of each pig were collected and stored in 2 mL centrifuge tubes containing 75% ethanol. The genomic DNA was extracted using phenol/chloroform extraction and stored at −20 °C [19]. The DNA concentration was measured with a Nanodrop 2000 spectrophotometer (Thermo Scientific, USA). All DNA samples were diluted to 30 ng/uL. Ratios of A260/A280 of these DNA samples were between 1.8–2.0. Only high-quality genomic DNA from all 655 samples was used for subsequent PCR reaction and genotyping.

The primers were designed using Premier 5.0 based on the VRTN sequence of GenBank (ID AB554652.1). The primers were synthesized in TSINGKE Company (Nanjing, China). The forward (F) primer sequence is 5′-GGCAGGGAAGGTGTTTGTTA-3′, and the reverse (R) primer sequence is 5′-GACTGGCCTCTGTCCCTTG-3′. PCR reactions were performed in a final volume of 25 μL, containing 22 μL 1.1 × T3 Super PCR Mix (Qingke, Nanjing, China), 1 μL genomic DNA: (30 ng/μL), 1 μL each of 10 nM/μL F and R primers. The PCR program was as follows: predenaturation at 96 °C for 2 min, 35 cycles of amplification at 96 °C for 20 s, 65 °C for 40 s, 72 °C for 45 s and a final extension step at 72 °C for 10 min. PCR products were separated by 2% agarose gel electrophoresis, and the genotypes were visually recorded according to the length of the amplicon. The allele (insertion) was represented by amplicons of 411 bp and the allele (deletion) by amplicons of 120 bp (Appendix A).

### 2.4. Statistical Analyses

The phenotypic variation, genotype frequencies, and allelic frequencies in two Suhuai pig populations were calculated using Microsoft Excel 2019. The heterozygosity (He), homozygosity (Ho), and the polymorphic information content (PIC) were estimated using Power Marker V3.0 software [20].

When estimating genetic correlations between RIB and carcass traits in SH-F, and genetic correlations between CBC and body size traits in SH-G, the fixed effects and covariate were determined based on the p value (Appendix A) of correlations between batch/sex/age and target traits using a general linear model (GLM) of SAS 9.2 software (SAS Institute Inc., Cary, NC, USA). When calculating genetic correlations of RIB and carcass traits in SH-F, for RIB, it was just considered for individuals as the random effect. When evaluating genetic correlations among CSL, CDL, and CWT in SH-F, batch and sex were considered as fixed factors, and age was considered as a covariate (Appendix A). When calculating genetic correlations among CC, AC, CBC, RC, BL, and BW, batch and sex were considered as fixed effects (Appendix A). The (co)variance components and their respective standard errors were estimated using DMU software based on multiple traits animal models [21]. Genetic correlation and phenotypic correlation were evaluated using formula A and B, respectively.
(1)$$\text{Formula A: Genetic correlation},\;r_{g}=\frac{cov\left(a_{1},a_{2}\right)}{\sqrt{\sigma_{a_{1}}^{2}\cdot \sigma_{a_{2}}^{2}}}$$
(2)$$\text{Formula B: Phenotypic correlation},\;r_{p}=\frac{cov\left(p_{1},p_{2}\right)}{\sqrt{\sigma_{p_{1}}^{2}\cdot \sigma_{p_{2}}^{2}}}$$

In formula A and B, $r_{g}$ is the coefficient of genetic correlation, and $r_{p}$ is the coefficient of phenotypic correlation; $cov\;\left(a_{1},\;a_{2}\right)$ is the additive genetic covariance of trait 1 and trait 2; $\sigma_{a_{1}}^{2}$ and $\sigma_{a_{2}}^{2}$ are the additive genetic variances of trait 1 and trait 2, respectively; $cov\;\left(p_{1},\;p_{2}\right)$ is the phenotypic covariance of trait 1 and trait 2; $\sigma_{p_{1}}^{2}$ and $\sigma_{p_{2}}^{2}$ are the phenotypic variances of trait 1 and trait 2, respectively.

Association analyses between the polymorphism of VRTN gene g.20311_20312ins291 and target traits in two Suhuai pig populations were conducted by mixed model of SAS 9.2 software (SAS Institute Inc., Cary, NC, USA).

We performed the association analyses between the polymorphism of VRTN gene g.20311_20312 ins291 and RIB in SH-F using model A. Association analyses between the polymorphism of this variant and CSL, CDL, and CWT in SH-F were conducted using model B. Association analyses between the polymorphism of this variant and six traits including CC, AC, CBC, RC, BL, and BW in SH-G were conducted using model C. We considered batch and sex as fixed effects in model B and C, and considered age as a covariate in model B, since they significantly affected carcass traits and body size traits in the correlation analyses (Appendix A).

(3)$$\text{Model A}:\;Y_{\mathrm{im}}=\mu +G_{i}+K_{m}+e_{\mathrm{im}}$$

(4)$$\text{Model B}:\;Y_{\mathrm{ijklm}}=\mu +G_{i}+B_{j}+S_{k}+D_{l}+K_{m}+e_{\mathrm{ijklm}}$$

(5)$$\text{Model C}:\;Y_{\mathrm{ijkm}}=\mu +G_{i}+B_{j}+S_{k}+K_{m}+e_{\mathrm{ijkm}}$$

In model A, B, and C, Y_im_, Y_ijklm_, and Y_ijkm_ are the vectors of the corresponding phenotypic value of the observed traits, μ is the mean of the phenotype of the observed traits, G_i_, B_j_, and S_k_ refer to the fixed effects of genotype, batch and sex, respectively. D_l_ represents the age, which is a covariate. Km is the random additive genetic effect, e_im_, e_ijklm_, and e_ijkm_ are residual errors.

## 3. Results

### 3.1. Descriptive Statistics for RIB, Carcass Traits, and Body Size/Body Weight Traits in Suhuai Pig Populations

The statistical description of RIB and carcass traits in 335 SH-F, and body size/body weight traits in 320 SH-G are presented in Table 1. In SH-F, these coefficients of variation of RIB, CSL, CDL, and CWT were 11.08%, 10.57%, 7.25%, and 6.00%, respectively. In SH-G, these coefficients of variation of CC, AC, CBC, RC, BL, and BW were 6.57%, 7.36%, 8.39%, 8.81%, 7.01%, and 12.68%, respectively.

### 3.2. Correlation Analyses of Carcass Traits in Suhuai Fattening Pigs and Body Size/Body Weight Traits in Suhuai Gilts

Table 2 displays the genetic correlations and phenotypic correlations between RIB and carcass traits in SH-F. Genetic correlation values of RIB with CSL, CDL, and CWT were 0.60, 0.58, and 0.09, respectively. Phenotypic correlation values of RIB with CSL, CDL, and CWT were 0.28, 0.25, and 0.04, respectively. The genetic correlation between CWT and CDL was the highest (0.95), and the genetic correlation between RIB and CWT was very low (0.09).

We analyzed genetic correlations and phenotypic correlations among body size/body weight traits in SH-G (Table 3). It showed that genetic correlation values of CBC with CC, AC, RC, BL, and BW were 0.64, 0.65, 0.89, 0.22, and 0.15, respectively, and phenotypic correlation values of CBC with CC, AC, RC, BL, and BW were 0.49, 0.48, 0.46, 0.43, and 0.40, respectively.

### 3.3. Genetic Parameters of the VRTN Gene g.20311_20312ins291 in Suhuai Pig Populations

As shown in Table 4, the ins/ins genotype frequencies were 0.3463, 0.3375, and 0.3420 in SH-F (335), SH-G (320), and T-Ps (655), respectively. Moreover, these genetic parameters (He, Ho, and PIC) of Suhuai pigs were estimated. The data showed that He of SH-F, SH-G, and T-Ps were 0.4823, 0.4887, and 0.4857, respectively. The ins allele frequencies were 0.5940, 0.5750, and 0.5847 in SH-F, SH-G, and T-Ps, respectively. Furthermore, the PIC of the *VRTN* gene g.20311_20312ins291 in SH-F, SH-G, and T-Ps were 0.3660, 0.3693, and 0.3677, respectively. The PIC belongs to medium polymorphic (0.25 < PIC < 0.5), indicating that Suhuai pigs had relatively plentiful genetic polymorphism of the *VRTN* gene g.20311_20312ins291.

### 3.4. Association Analyses of the Polymorphism of VRTN Gene g.20311_20312ins291 with RIB and Carcass Traits in Suhuai Fattening Pigs and Body Size/Body Weight Traits in Suhuai Gilts

Table 5 shows the results of the association analyses between the polymorphism of the *VRTN* gene g.20311_20312ins291 and RIB, carcass traits, and body size traits in Suhuai pigs. It showed that the polymorphism of the *VRTN* gene g.20311_20312ins291 was extremely significantly associated with RIB (*p =* 0.00) and CDL (*p =* 0.00), and there was a tendency of association between this variant and CSL (*p =* 0.06) in SH-F. In SH-G, the polymorphism of the *VRTN* gene g.20311_20312ins291 was significantly associated with CBC (*p* = 0.04), ins/ins genotype was the advantageous genotype, and there was a tendency of association between this variant and RC (*p* = 0.07). However, the *VRTN* gene g.20311_20312ins291 had no influence on CWT in SH-F, and CC, AC, BL, and BW in SH-G.

The RIB of individuals (31.00 ± 1.21) with ins/ins genotype were significantly higher than individuals (30.46 ± 1.08) with ins/del genotype and individuals (29.55 ± 1.15) with del/del genotype (*p* < 0.01), respectively. The CDL of individuals (75.60 ± 1.27) with ins/ins genotype were significantly longer than individuals (74.50 ± 1.23) with ins/del genotype (*p* < 0.05) and longer than individuals (74.00 ± 1.40) with del/del genotype (*p* < 0.01), respectively.

From these data, we found that the *VRTN* gene g.20311_20312ins291 had an additive effect (ins/ins > ins/del > del/del) on RIB, CDL, and CBC in Suhuai pig populations.

## 4. Discussion

Several studies have demonstrated that an increase in vertebral numbers has an influence on the improvement of pork production. Therefore, our study measured RIB, carcass traits in 335 SH-F, and body size traits in 320 SH-G. There were phenotypic variations of RIB, CSL, CDL, CWT, CC, AC, CBC, RC, BL, and BW in Suhuai pig populations, perhaps due to Suhuai pig being a new breed which was cultivated by Huai pigs (with a low number of ribs and small body size) and Large White (with a high number of ribs and large body size). There were positive genetic correlations between RIB and CSL, as well as between RIB and CDL in SH-F. These results were similar to the previous study, in which the value was 0.56 in Duroc [22]. The genetic correlation between CBC and BL in SH-G was consistent with some studies, in which these correlations were 0.305 and 0.552 in Large White and American Duroc, respectively [23,24]. Furthermore, the result of the low correlation between RIB and CWT was similar to previous studies, in which the negative genetic correlation (−0.12) and phenotypic correlation (−0.06) between RIB and CWT were found in Duroc [22]. The correlation between RIB and CWT was negative in PIC pigs [25]. These results implied that RIB had no influence on CWT; however, this result should be confirmed further using other populations.

Prior studies found that the frequency of ins/ins genotype were 0.70 (*n* = 764) and 0.43 (*n* = 583) in Landrace and Large White, respectively [26]. In our data, we found the *VRTN* gene g.20311_20312ins291 was polymorphic, and there were three genotypes in Suhuai pigs. The ins/ins genotype frequency were 0.3463 (*n* = 335), and 0.3375 (*n* = 320) in SH-F and SH-G, respectively. The ins/ins genotype frequency of Suhuai pigs was slightly lower than Landrace and Large White. These results may be due to the genealogy of Suhuai pigs, which contains approximately 75% Large White ancestry. Moreover, Suhuai pigs did not strongly select for body size, resulting in relatively low ins/ins frequencies. These results implied that there was a certain space for the breeding and improvement of Suhuai pigs.

The *VRTN* gene g.20311_20312ins291 was significantly associated with TVN in Western commercial populations and Sutai pigs [11]. The polymorphism of the *VRTN* gene g.20311_20312ins291 was also significantly associated with TVN and CL in F_2_ intercross of Erhualian and White Duroc [26]. Moreover, the *VRTN* gene g.20311_20312ins291 was significantly associated with RIB in the Tongcheng × Large White crossbreed population (*n* = 448) [27]. These results were consistent with SH-F, in which the polymorphism of the *VRTN* gene g.20311_20312ins291 was associated with RIB and CDL in SH-F. In contrast, the polymorphism of the *VRTN* gene g.20311_20312ins291 did not affect BL in SH-G; this is in keeping with Yang’s study, in which they claimed this variant did not influence BL of Landrace [26]. On the one hand, it is possible that variants on LVN and TVN determine the BL. On the other hand, there may be other candidate genes including *Hox A*, *Hox B*, *LTBP2,* and *TGFβ3* that affect the vertebral numbers of pigs [13,14,15,16].

The RIB of individuals with ins/ins genotype have 1.45 more ribs than individuals with del/del genotype in SH-F. This result was very similar to Fan’s study in Sutai pigs, in which individuals (15.07 ± 0.13) with ins/ins genotype had 1.19 more ribs than individuals (13.88 ± 0.19) with del/del genotype [11]. Moreover, Fan et al. showed that the polymorphism of the *VRTN* gene was significantly associated with TVN of DLL (*p <* 0.01), and individuals (15.63 ± 0.41) with ins/ins genotype have 0.92 more TVN than individuals (14.71 ± 0.52) with del/del genotype. These results implied that the *VRTN* gene g.20311_20312ins291 could be one factor influencing the RIB and CDL in Suhuai pigs. Of course, there may be other genes that affect the variation of vertebral numbers.

For CBC, a study showed that the *VRTN* genotypes (W/W, W/Q, and Q/Q) had no influence on CBC in Duroc gilts, boar, and barrow [28]. Our study found that the polymorphism of the *VRTN* gene g.20311_20312ins291 was significantly associated with CBC in SH-G. To our knowledge, our study is the first to show that the *VRTN* gene, located on SSC7, may be a candidate gene affecting the CBC of Chinese Suhuai pigs. Furthermore, there have been many studies that have demonstrated QTLs related to CBC on SSC7. For instance, there were multiple significant single nucleotide polymorphisms (SNPs) that were associated with CBC, and these QTLs were mainly located on SSC7 (from 35.0 Mb to 43.9 Mb) [29,30], and 138 SNPs were significantly associated with BL and CBC in Large White × Min pig F_2_ resource population [31]. Additionally, the most significant SNP was MARC0058766 (at 34.80 Mb) on SSC7 for CBC, which is close to *HMGA1* (high mobility group AT-Hook 1) gene, and the distance between *HMGA1* gene and *VRTN* gene is approximately 67.29 Mb [32].

CBC is one of the important appearance traits in the pig breeding program. The large CBC could enhance the capacity to bear excessive body weight and to resist injuries. A large CBC is in demand for bearing the bodyweight when RIB and CWT were increased due to breeding. Interestingly, our study found that the ins/ins genotype of the *VRTN* gene g.20311_20312ins291 can increase both RIB, CDL, and CBC. We also found that positive genetic correlations and phenotypic correlations between RIB and (CSL, CDL, and CWT) in SH-F and the positive genetic correlations and phenotypic correlations between CBC and (BL, BW) in SH-G. These results imply that the *VRTN* gene g.20311_20312ins291 could be used as a potential marker to select RIB, CDL, and CBC in Suhuai pigs at the same time.

## 5. Conclusions

There were phenotypic variations of RIB, CSL, CDL, CWT, CC, AC, CBC, RC, BL, and BW in Suhuai pig populations. Phenotypic and genetic correlations between RIB and CSL, CDL, and CWT were positive in SH-F. Phenotypic and genetic correlations between CBC and BL and BW were positive in SH-G. The polymorphism of the *VRTN* gene g.20311_20312ins291 was associated with RIB and CDL in SH-F and CBC in SH-G. This variant could be used as a potential molecular marker to improve RIB, CDL, and CBC in Suhuai pig breeding.

## Figures and Tables

**Table 1 animals-10-00484-t001:** Descriptive statistics analyses of the number of ribs and carcass traits in Suhuai fattening pigs and body size/body weight traits in Suhuai gilts.

Populations	Traits	Numbers	Ranges	Mean ± SE	CVs, %	Medians
SH-F	RIB	335	28.00–34.00	30.51 ± 0.13	11.08	30.00
	CSL (cm)		69.00–108.00	88.63 ± 0.32	10.57	88.20
	CDL (cm)		58.00–93.00	75.03 ± 0.29	7.25	75.00
	CWT (kg)		40.05–84.40	55.96 ± 0.19	6.00	59.23
SH-G	CC (cm)	320	75.00–108.00	89.80 ± 0.27	6.57	88.70
	AC (cm)		80.00–120.00	101.16 ± 0.34	7.36	100.00
	CBC (cm)		11.00–20.00	14.67 ± 0.06	8.39	14.50
	RC (cm)		50.00–90.00	72.67 ± 0.30	8.81	72.00
	BL (cm)		70.00–115.00	96.16 ± 0.31	7.01	95.00
	BW (kg)		36.80–85.00	57.81 ± 0.35	12.68	57.60

Note: RIB, number of ribs; CSL, carcass straight length; CDL, carcass diagonal length; CWT, carcass weight; CC, chest circumference; AC, abdominal circumference; CBC, cannon bone circumference; RC, rump circumference; BL, body length; BW, body weight; CVs, coefficient of variations; mean ± SE, the means with standard errors for traits; SH-F, Suhuai fattening pigs; SH-G, Suhuai gilts.

**Table 2 animals-10-00484-t002:** Correlation analyses between RIB and carcass traits in Suhuai fattening pigs.

Traits	RIB	CSL	CDL	CWT
RIB		0.60 ** (0.25)	0.58 ** (0.29)	0.09 (0.35)
CSL	0.28 **		0.83 ** (0.12)	0.68 ** (0.17)
CDL	0.25 **	0.71 **		0.95 ** (0.12)
CWT	0.04	0.51 **	0.53 **	

Note: Values above the diagonal are genetic correlations among the traits; values below the diagonal are phenotypic correlations among the traits. Standard errors are shown in parentheses. ** represents *p* < 0.01, * represents *p* < 0.05. RIB, number of ribs; CSL, carcass straight length; CDL, carcass diagonal length; CWT, carcass weight.

**Table 3 animals-10-00484-t003:** Correlation analyses between body size and body weight in Suhuai gilts.

Traits	CC	CBC	AC	RC	BL	BW
CC		0.64 ** (0.13)	0.98 ** (0.03)	0.63 ** (0.12)	0.92 ** (0.08)	0.93 ** (0.05)
CBC	0.49 **		0.65 ** (0.15)	0.89 ** (0.08)	0.22 ** (0.19)	0.15 * (0.17)
AC	0.80 **	0.48 **		0.61 ** (0.15)	0.94 ** (0.06)	0.89 ** (0.07)
RC	0.49 **	0.46 **	0.42 **		0.59 ** (0.16)	0.25 ** (0.14)
BL	0.64 **	0.43 **	0.64 **	0.41 **		0.82 ** (0.09)
BW	0.72 **	0.40 **	0.68 **	0.33 **	0.60 **	

Note: Values above the diagonal are genetic correlations among the traits; values below the diagonal are phenotypic correlations among the traits. Standard errors are shown in parentheses. ** represents *p* < 0.01, * represents *p* < 0.05. CC, chest circumference; CBC, cannon bone circumference; AC, abdominal circumference; RC, rump circumference; BL, body length; BW, body weight.

**Table 4 animals-10-00484-t004:** Genetic parameters of the *VRTN* gene g.20311_20312ins291 in Suhuai pig populations.

Groups	Numbers	Genotype Frequencies			Allele Frequencies		Ho	He	Ne	PIC
		ins/ins	ins/del	del/del	ins	del				
SH-F	335	0.3463 (116)	0.4955 (166)	0.1582 (53)	0.5940	0.4060	0.5177	0.4823	1.9317	0.3660
SH-G	320	0.3375 (108)	0.4750 (152)	0.1875 (60)	0.5750	0.4250	0.5113	0.4887	1.9560	0.3693
T-Ps	655	0.3420 (224)	0.4855 (318)	0.1725 (113)	0.5847	0.4153	0.5143	0.4857	1.9442	0.3677

Note: Ho, homozygosity; He, heterozygosity; Ne, effective allele numbers; PIC, polymorphic information content; SH-F, Suhuai fattening pigs; SH-G, Suhuai gilts; T-Ps, total populations; ins, insertion; del, deletion.

**Table 5 animals-10-00484-t005:** Association analyses of the polymorphism of the *VRTN* gene g.20311_20312ins291 with RIB and carcass traits in Suhuai fattening pigs and body size/body weight traits in Suhuai gilts.

Populations	Traits	*VRTN* Genotypes			*p* Values
		ins/ins	ins/del	del/del	
SH-F		(*n* = 116)	(*n* = 166)	(*n* = 53)	
	RIB	31.00 ± 1.21 ^A^	30.46 ± 1.08 ^B^	29.55 ± 1.15 ^C^	0.00
	CSL (cm)	88.99 ± 1.30	88.17 ± 1.25	87.93 ± 1.45	0.06
	CDL (cm)	75.60 ± 1.27 ^A,a^	74.50 ± 1.23 ^A,B,b^	74.00 ± 1.40 ^B,c^	0.00
	CWT (kg)	61.77 ± 1.56	59.63 ± 1.26	59.06 ± 1.10	0.57
SH-G		(*n* = 108)	(*n* = 152)	(*n* = 60)	
	CC (cm)	89.27 ± 0.70	88.22 ± 0.62	88.21 ± 0.83	0.26
	AC (cm)	100.21 ± 0.76	99.63 ± 0.60	99.10 ± 0.92	0.50
	CBC (cm)	15.38 ± 0.13	15.12 ± 0.11	14.95 ± 0.15	0.04
	RC (cm)	73.38 ± 0.58	72.66 ± 0.50	71.32 ± 0.71	0.07
	BL (cm)	94.60 ± 0.71	94.98 ± 0.63	93.81 ± 0.87	0.41
	BW (kg)	57.03 ± 0.89	56.28 ± 0.80	55.35 ± 1.07	0.32

Note: ^a,b,c^ values with different superscripts within the same row in a particular population differ significantly at *p* < 0.05. ^A,B,C^ values with different superscripts within the same row in a particular population differ significantly at *p* < 0.01. RIB, number of ribs; CSL, carcass straight length; CDL, carcass diagonal length; CWT, carcass weight; CC, chest circumference; AC, abdominal circumference; CBC, cannon bone circumference; RC, rump circumference, BL, body length; BW, body weight. SH-F, Suhuai fattening pigs; SH-G, Suhuai gilts; ins, insertion; del, deletion.

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
