# Peer review of "Polymorphism of VRTN Gene g.20311_20312ins291 Was Associated with the Number of Ribs, Carcass Diagonal Length and Cannon Bone Circumference in Suhuai Pigs"

_animals, 2020, doi:10.3390/ani10030484_

Round 1
Reviewer 1 Report
The paper reports the results of an interesting research in which a gene polimorphism is associated to a trait of economic interest in pig breeding.
Although the data are interesting, their description is lacking. For the purpose of publication, several corrections must be made.
In particular the English language style must be improved throughout the article.
Major revisions:
1) please rewrite Discussion section.
Some of the data reported in this section should be moved to Introduction section (for example LINE 239-245 and LINE 280-LINE 288). The section needs a wide revision as regard the english style.
2) Please improve Conclusion section. It is very concise and it needs a revision as regard the english style.
3) check the following sentences:
LINE 17-19; LINE 74-80; LINE 113-115; Line 129-133; LINE 142-148; .
Minor Revisions:
LINE 23: …in “Suhuai pigs” could be changed with “… this breed”
LINE25: Plese change “ …was the advantageous genotype for…” with “…was advantageous for…”
LINE 30: Please change “…polymorphism in two populations.” with “…polymorphism in both populations.”
LINE 42-43: Plese change “Carcass traits, such as carcass weight and carcass length, are important economic traits in meat production.” with “Carcass traits, such as weight and length, are of economic importance in meat production.”
LINE 44: Please change “…cannon bone circumference and so on, are important goals of….” with “…cannon bone circumference, are goals pursued in….”
LINE 46: please delete “which up to”
LINE 48-49: Please change “…pigs consist of five parts: cervical vertebrae,thoracic vertebrae, lumbar vertebrae, sacral vertebrae and caudal vertebrae.” with “…pigs are classified in five types: cervical, thoracic, lumbar, sacral and caudal vertebrae.”
LINE 51-52 Please change “…in the thoracic vertebral number (TVN) and lumbar vertebral number (LVN). The TVN ranged from 13 to 17 and the LVN ranged….” with “… in their number. The thoracic vertebral number (TVN) ranged from 13 to 17 and the Lumbar vertebral number (LVN) ranged… ” .
LINE 57: Change after with thanks to
LINE 59: Please change “…pork carcass, RIB is…” with “…pork carcass , number of ribs (RIB)…”
LINE 92: Please change “All experimental animals were performed…” with “All experiments were performed…”
LINE 108: please describe the protocol used for DNA extraction (including reagents and their manufacturer) in the text or in supplementary material or cite a reference of the used protocol.
LINE 110: Please change “…and the ratio of light….” with “… and had a ratio of…”
Specify the accession number of the sequence used to generate the primers and the software used.
Reference 3 is incomplete. Reference 15 seems not to be related to the sentence.
Reviewer 2 Report
The major concern is the association analysis. 1) As parent genetics is expected to have a major impact on the traits investigated, why this was not corrected in the statistical models? 2) Why RIB was not corrected for gender as other traits? 3) Why the body size traits were not corrected for age, despite it may not be huge different in those Suhuai pilts?
Other comments:
The SNP should be described following the HGVS nomenclature (http://varnomen.hgvs.org/).
The PCR reagents should be given with the working concentrations and not the recipe.
Round 2
Reviewer 2 Report
The authors claimed that age was not found to have any effect on all of the body size and body weight traits, which is very difficult to understand. The absence of association between age and body size/weight traits suggests that phenotypic data or the statistical analysis may be incorrect, as it is expected that adult pigs will have bigger body and heavy weight than younger pigs. This needs to be carefully checked and validated, as it may significantly affect the outcome of the results.
The use of mutation in this manuscript is inappropriate, as "mutation" is defined as any change in a DNA sequence away from normal. This implies there is a normal allele that is prevalent in the population and that the mutation changes the normal allele to a rare and abnormal variant. Actually you don't know which is normal allele and which is abnormal allele regarding the variation that you described in this study. I would suggest use variation or polymorphism to replace mutation.
